# Measuring ligand efficacy at the mu-opioid receptor using a conformational biosensor

**Kathryn E Livingston[1,2], Jacob P Mahoney[1,2], Aashish Manglik[3], Roger K Sunahara[4], John R Traynor[1,2]***

[1]Department of Pharmacology, University of Michigan Medical School, Ann Arbor, United States; [2]Edward F Domino Research Center, University of Michigan, Ann Arbor, United States; [3]Department of Pharmaceutical Chemistry, School of Pharmacy, University of California San Francisco, San Francisco, United States; [4]Department of Pharmacology, University of California San Diego School of Medicine, La Jolla, United States

**Abstract** The intrinsic efficacy of orthosteric ligands acting at G-protein-coupled receptors (GPCRs) reflects their ability to stabilize active receptor states (R*) and is a major determinant of their physiological effects. Here, we present a direct way to quantify the efficacy of ligands by measuring the binding of a R*-specific biosensor to purified receptor employing interferometry. As an example, we use the mu-opioid receptor (μ-OR), a prototypic class A GPCR, and its active state sensor, nanobody-39 (Nb39). We demonstrate that ligands vary in their ability to recruit Nb39 to μ-OR and describe methadone, loperamide, and PZM21 as ligands that support unique R* conformation(s) of μ-OR. We further show that positive allosteric modulators of μ-OR promote formation of R* in addition to enhancing promotion by orthosteric agonists. Finally, we demonstrate that the technique can be utilized with heterotrimeric G protein. The method is cell-free, signal transduction-independent and is generally applicable to GPCRs.
DOI: https://doi.org/10.7554/eLife.32499.001

***For correspondence:**
jtraynor@umich.edu

**Competing interests:** The authors declare that no competing interests exist.

## Introduction

The intrinsic efficacy of a ligand acting at a 7-transmembrane (TM) domain, G-protein-coupled receptor (GPCR) describes its ability to shift the equilibrium from inactive receptor (R) in favor of active receptor states (R*). Thus, intrinsic efficacy is the parameter that distinguishes agonists, partial agonists, antagonists, and inverse agonists and is a major determinant of pharmacological activity of a drug molecule, along with receptor affinity. This is particularly highlighted by the mu-opioid receptor (μ-OR), a GPCR responsible for both the pain relieving and unwanted effects of opioid drugs, such as life-threatening respiratory depression and the rewarding properties that underlie addiction to drugs such as morphine, oxycodone, and heroin. Even though opioid drugs bind to the same orthosteric site on μ-OR, the physiological outcomes observed are determined by their degree of intrinsic efficacy and the efficacy requirements of the system (*Morgan and Christie, 2011*). For example, the discriminative stimulus of opioid agonists varies depending on efficacy (*Walker et al., 2004*), suggesting that opioid agonist efficacy is a determinant of abuse potential. Thus, it has been shown that higher efficacy ligands have greater addictive liability compared to lower-efficacy opioid ligands (*Center for Substance Abuse Treatment, 2004*). In order to be able to predict the ability of μ-OR agonists to cause various physiological effects, an understanding of their intrinsic efficacy is required.

Current approaches to determine the intrinsic efficacy of agonists rely upon measurement of signaling downstream of the receptor in tissue- or cell-based systems. In spite of the use of methods to account for signal amplification which can cause system and measurement bias the calculated intrinsic efficacy of ligands can vary based on the species of the tissue or cellular background, ligand bias, and temporal effects (*Kenakin, 2002*; *Kenakin and Christopoulos, 2011*; *Luttrell and Kenakin, 2011*; *Klein Herenbrink et al., 2016*). For example, using the same cellular background, the putatively biased opioid ligand TRV130 initiates arrestin recruitment (74% of morphine) downstream of mouse μ-OR, less with human μ-OR (14% of morphine), and shows undetectable levels using rat μ-OR (*DeWire et al., 2013*). Complications such as these make correlations of physiological effects to values of efficacy and 'bias factors' (*Rajagopal et al., 2010*; *Kenakin et al., 2012*) difficult to interpret. Therefore, in this study we sought to establish a method to evaluate the intrinsic efficacy of opioid ligands utilizing a cell-free assay, thereby removing confounds of signaling outputs and signal amplification.

The recently solved crystal structure of μ-OR in complex with the highly efficacious agonist BU72 utilized nanobody 39 (Nb39), a camelid antibody fragment, to stabilize active μ-OR (*Huang et al., 2015*). Nb39 enhances agonist affinity at μ-OR and stabilizes conformational changes in the receptor associated with a signaling-competent, active-like state, including an outward movement of TM6 (*Farrens et al., 1996*; *Palczewski et al., 2000*; *Rasmussen et al., 2011b*). Nanobodies are small, monomeric proteins that can be utilized as conformational biosensors and have therefore been used as tools to monitor formation of active-state β2-adrenergic receptors in live cells (*Irannejad et al., 2013*). Consequently, we sought to use Nb39 as a G protein mimic and a probe to detect active-state conformation of purified monomeric μ-OR in reconstituted high density lipoprotein (rHDL) particles, comprised of the lipids 1-palmitoyl-2-oleoyl-sn-glycero-3-phosphocholine and 1-palmitoyl-2-oleoyl-sn-glycero-3-phosphoglycerol and Apolipoprotein-A1 (Apo-A1) (*Kuszak et al., 2009*), using a variety of orthosteric ligands. We predicted that the intrinsic efficacy of an agonist will determine the rate of Nb39 binding to μ-OR.

We also wished to utilize Nb39 to probe the mechanism by which small molecule positive allosteric modulators (PAMs) alter the activity of orthosteric μ-OR ligands. μ-PAMS, exemplified by BMS-986122 (*Burford et al., 2013*), enhance the affinity and/or efficacy of orthosteric ligands for μ-OR, but show a distinct dependence on which orthosteric ligand is used to probe the allosteric interaction, with the degree of cooperativity being sensitive to the intrinsic efficacy of the orthosteric ligand (*Burford et al., 2013*; *Livingston and Traynor, 2014*). We have suggested that the mechanism of allosteric enhancement of orthosteric agonist activity involves a disruption of binding of the endogenous negative allosteric modulator Na$^+$, thereby stabilizing an agonist-bound active (R*) state of the μ-OR (*Livingston and Traynor, 2014*).

In this report, we demonstrate that the rate of Nb39 binding to purified μ-OR simulates the action of heterotrimeric G protein and provides a measure of the intrinsic efficacy of opioid ligands, and that the efficacy rankings determined using this second messenger- and cell/tissue-independent assay are significantly correlated with other measures of efficacy. Moreover, we observed differences in Nb39 dissociation suggestive of ligand-specific R* states. Finally, we show that μ-PAMs can alone promote Nb39 binding, indicating that they stabilize formation of R* to increase the apparent efficacy of partial agonist drugs such as morphine.

## Results

### Measure of orthosteric agonist efficacy using an interferometry-based technique

Nb39 enhances the affinity of orthosteric agonists, such as BU72, to bind μ-OR (*Huang et al., 2015*) by stabilizing active (R*) states of μ-OR. Since agonists shift the equilibrium of receptor to R* in proportion to their efficacy to activate downstream signaling, we predicted that agonists should enhance the binding of Nb39 in an efficacy-dependent manner. To test this hypothesis, we implemented an interferometry-based technique to study the association and dissociation kinetics of Nb39 binding to μ-OR in rHDL. The formation of μ-OR containing rHDL was performed using a low μ-OR to rHDL ratio and an ApoAI construct that heavily favors incorporation of μ-OR monomers into rHDL particles (*Kuszak et al., 2009*). The μ-OR-containing rHDL particles were immobilized on an

interferometry probe, and the probe was then exposed to saturating concentrations of ligands and a sub-saturating concentration of Nb39 (1 µM) (*Figure 1*). As shown in *Figure 2*, there was no detectable binding of Nb39 to µ-OR in the absence of ligand, indicating a lack of spontaneous formation of active µ-OR, even with Nb39 present. This supports published research indicating low levels of constitutive activity of µ-OR (*Divin et al., 2009*; *Connor and Traynor, 2010*).

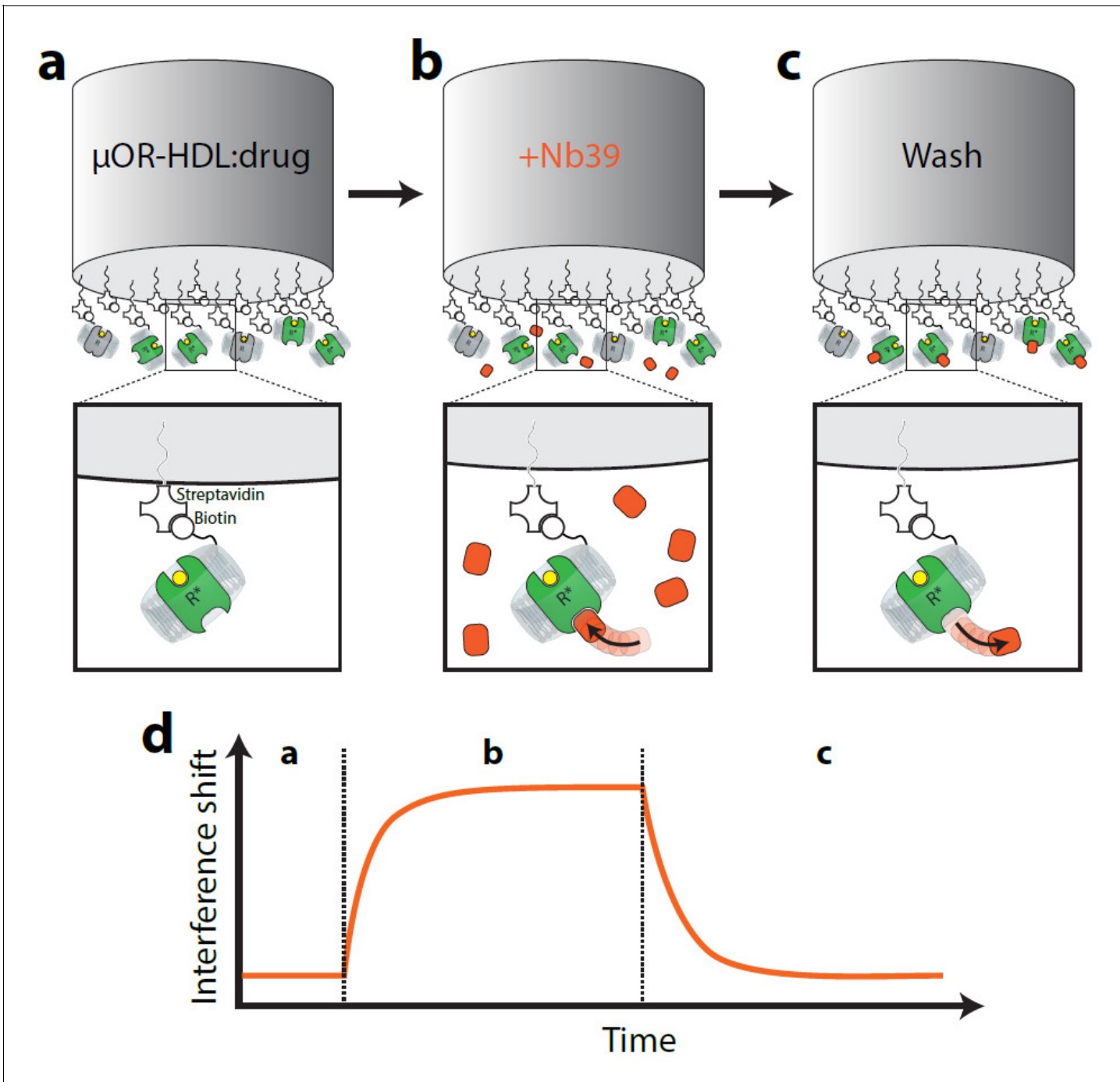

**Figure 1.** Cartoon schematic of interferometry assay to detect µ-OR:Nb39 interactions. Briefly, (a) Biotin-conjugated rHDL particles containing µ-OR are loaded on streptavidin-coated tips and incubated with saturating ligand (or vehicle) for 10 min. (b) Probe is exposed to Nb39 for five min in the presence of ligand or vehicle until steady state is reached. (c) In the presence of ligand, probe is moved to well containing no Nb39 to monitor dissociation for 5 min.

DOI: https://doi.org/10.7554/eLife.32499.002

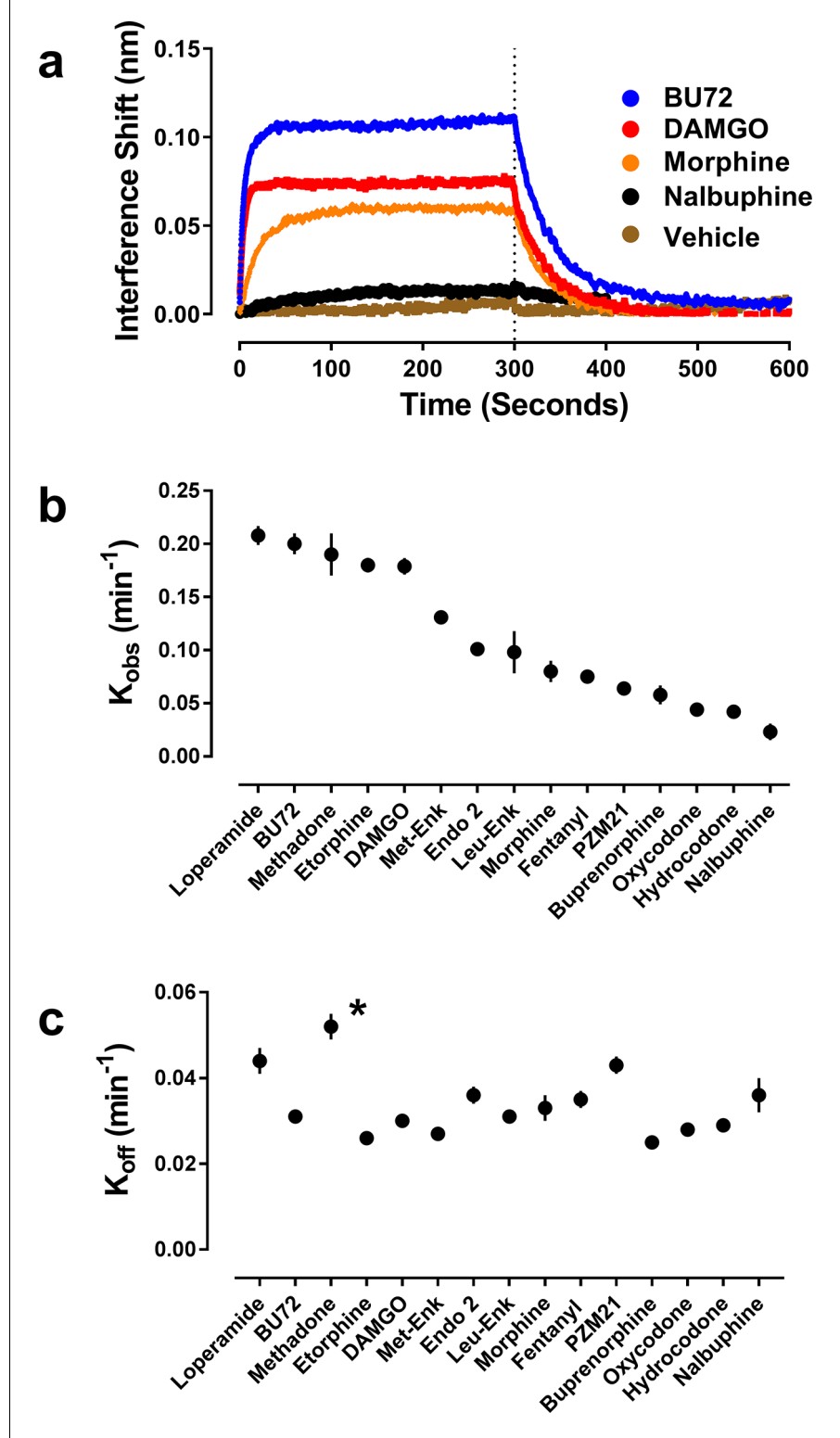

**Figure 2.** Orthosteric ligand-mediated Nb39 association and dissociation in μ-OR-rHDL. As described in the Materials and methods, the association and dissociation of Nb39 (1 μM) was measured using an OctetRed® instrument. Shown is a representative experiment comparing four orthosteric agonists at 30 μM (a) Using GraphPad Prism 6.02, one-phase association lines were fit and the calculated $k_{obs}$ (b) and $k_{off}$ (c) for each ligand are plotted (±s.e.m.). Rate constants are means of multiple independent experiments as listed in see

*Figure 2 continued on next page*

*Figure 2 continued*

*Table 1*. *Dissociation of Nb39 from the methadone-bound receptor is statistically different from all other ligands (one-way ANOVA with Tukey's post-hoc test).
DOI: https://doi.org/10.7554/eLife.32499.003

Conversely, pre-incubation of μ-OR with a wide range of agonists of varying structure, both peptides and small molecules, caused binding of Nb39, although to varying degrees (*Table 1*). In particular, the presence of a saturating concentration (30 μM) of the high-efficacy agonist BU72 (*Neilan et al., 2004*; *Huang et al., 2015*) drove robust and rapid binding of Nb39 (*Figure 2*, *Table 1*). In the presence of the high-efficacy peptide agonist DAMGO, Nb39 bound to μ-OR with a similar rate constant, but DAMGO lead to less overall binding of Nb39 compared to BU72. In contrast, pre-incubation with the partial agonist morphine caused slower Nb39 association and less overall binding of Nb39 relative to BU72 and DAMGO (*Figure 2*, *Table 1*), and the rate constant for Nb39 binding in the presence of the low-efficacy nalbuphine was slower still. Neither of the orthosteric antagonists (naloxone or diprenorphine) promoted a μ-OR:Nb39 interaction (*Table 1*).

The above data indicate that the binding of Nb39 is related to agonist efficacy, with higher efficacy ligands producing an increased Nb39 binding signal and a more rapid association of Nb39. However, the magnitude of the interference shift is problematic to use as the readout for efficacy since this depends on the initial amount of stable receptor loaded onto the probe which is difficult to control because of problems of equalizing receptor loads onto the probe, use of different receptor preparations, plus the fact that is not known how much receptor is actually functional.

**Table 1.** Association and dissociation kinetics of Nb39 to μ-OR-rHDL in the presence of various agonists.

$k_{obs}$ and $k_{off}$ were determined for each independent experiment (number of individual experiments indicated in 'n' column) and averaged. One-phase association and single-phase exponential decay models were used. Half-time values ($t_{1/2}$) numbers were calculated from the respective k values ($t_{1/2}$ = 0.693/k). A one-way ANOVA was performed followed by a Tukey post-hoc test. Methadone was found to be statistically different compared to all other orthosteric ligands other than loperamide and PZM21. Both loperamide and PZM21 were also found to be statistically different from several other ligands, though not as many as methadone.

| Ligand | $k_{obs}$ ± SEM (min$^{-1}$) | $t_{1/2}$ass (sec) | $k_{off}$ (min$^{-1}$) | $t_{1/2}$dis (sec) | N |
|---|---|---|---|---|---|
| BU72 | 0.20 ± 0.01 | 3.5 | 0.031 ± 0.001 | 22 | 14 |
| DAMGO | 0.179 ± 0.008 | 3.9 | 0.030 ± 0.001 | 23 | 6 |
| Leu-Enk | 0.098 ± 0.02 | 7.1 | 0.031 ± 0.001 | 23 | 9 |
| L-Methadone | 0.19 ± 0.02 | 3.6 | 0.052 ± 0.003 | 13 | 6 |
| Morphine | 0.08 ± 0.01 | 8.5 | 0.033 ± 0.003 | 21 | 7 |
| Nalbuphine | 0.023 ± 0.008 | 30 | 0.036 ± 0.004 | 19 | 11 |
| PZM21 | 0.064 ± 0.004 | 11 | 0.043 ± 0.002 | 16 | 6 |
| Endomorphin 2 | 0.101 ± 0.002 | 6.9 | 0.036 ± 0.002 | 19 | 6 |
| Loperamide | 0.208 ± 0.009 | 3.2 | 0.044 ± 0.003 | 16 | 6 |
| Oxycodone | 0.044 ± 0.001 | 16 | 0.028 ± 0.001 | 25 | 9 |
| Etorphine | 0.180 ± 0.007 | 3.9 | 0.026 ± 0.001 | 27 | 6 |
| Fentanyl | 0.075 ± 0.005 | 9.2 | 0.035 ± 0.002 | 20 | 6 |
| Met-Enk | 0.131 ± 0.004 | 5.3 | 0.027 ± 0.001 | 26 | 6 |
| Hydrocodone | 0.042 ± 0.004 | 17 | 0.029 ± 0.001 | 24 | 6 |
| Buprenorphine | 0.058 ± 0.009 | 12 | 0.025 ± 0.001 | 27 | 6 |
| Naloxone | n/a | —— | n/a | —— | 3 |
| Diprenorphine | n/a | —— | n/a | —— | 3 |
| BMS-986122 | 0.012 ± 0.001 | 56 | 0.027 ± 0.003 | 25 | 10 |
| BMS-986187 | 0.025 ± 0.006 | 28 | 0.037 ± 0.005 | 19 | 11 |

DOI: https://doi.org/10.7554/eLife.32499.004

Consequently, we focused on the association rate constants ($K_{obs}$ and $t_{1/2}$). First, we made sure that agonist binding kinetics was not a confounding factor in measuring Nb39 binding. Opioid ligands have very fast binding kinetics (*Huang et al., 2015*) and to confirm equilibrium was reached we used maximal agonist concentrations and for two distinct ligands, DAMGO and PMZ21 showed these gave the same results with 10 or 30 min incubation (PZM21 $k_{obs}$ (10 min)=0.060 ± 0.008 $min^{-1}$ and $k_{obs}$ (30 min)=0.064 ± 0.004 $min^{-1}$; DAMGO $k_{obs}$ (10 min)=0.15 ± 0.006 $min^{-1}$ and $k_{obs}$ (30 min) =0.16 ± 0.008 $min^{-1}$). Then, to confirm that the rate constant of Nb39 association to agonist-bound receptor accurately reflects the intrinsic efficacy of a given agonist, we compared Nb39 association with four accepted methods of agonist efficacy measurement: (i) maximal ability to activate hetero-trimeric G protein (*Strange, 2008*), (ii) intrinsic efficacy as defined by Ehlert's equation (*Ehlert, 1985*), (iii) reduction in agonist affinity in the presence of $Na^+$ ions and GTP (*Lee et al., 1999*; *Zhen et al., 2015*) and (iv) tau (τ) as calculated using the Black-Leff operational model (*Black and Leff, 1983*). These complimentary methods of agonist efficacy determination generally agree with one another, but use different measurements of receptor activity to calculate efficacy.

The Nb39 association half-time ($t_{1/2}$) (*Table 1*) correlated with the ability of each orthosteric ligand, when used at a saturating concentration (10 µM), to stimulate G protein activation as measured by GTPγ$^{35}$S binding (taken from [*Livingston and Traynor, 2014*]), giving $r^2 = 0.75$, p<0.0001 (*Figure 3a*). Next, we compared the Nb39 association data with the intrinsic efficacy (e) of the various orthosteric ligands, calculated using the Ehlert equation (*Ehlert, 1985*) with potency and maximal response obtained from GTPγ$^{35}$S-binding assays and affinity values obtained using radioligand competition binding (*Livingston and Traynor, 2014*). There was a statistically significant correlation between the $t_{1/2}$ of Nb39 binding and intrinsic efficacy ($r^2 = 0.44$, p=0.02; *Figure 3b*). For instance, etorphine and BU72 are high-efficacy ligands with equivalent e values (4.7) and pre-incubating µ-OR with either ligand results in a similar Nb39 association half-time ($t_{1/2} = 3.5$ sec for BU72, 3.9 sec for etorphine). In this case, the correlation was weaker than the comparison between Nb39 association and GTPγ$^{35}$S maximum stimulation, and we excluded data collected with nalbuphine, as a potency value could not be determined due to its weak activation of G protein in our system.

Determination of efficacy can vary based on signaling output chosen, especially in the case of bias where ligands may preferentially activate certain pathways over others. To avoid the use of a signaling measure, that is G-protein activation or β-arrestin recruitment, we examined the shift in agonist affinity in response to the presence of $Na^+$ ions and GTP, using GTPγS. It is known that addition of $Na^+$ ions and guanine nucleotide decreases the affinity of agonists to bind µ-OR and that the degree of shift is larger for higher efficacy ligands (*Lee et al., 1999*). Using our previously published data (*Livingston and Traynor, 2014*), we plotted the shift in affinity of the orthosteric ligands by the addition of NaCl/GTPγS (100 mM and 10 µM, respectively) versus the calculated $t_{1/2}$ of Nb39 association to ligand-bound µ-OR, from *Table 1*. This correlation was significant (*Figure 3c*, $r^2 = 0.73$, p=0.002), although BU72 and etorphine had to be excluded from the analysis due to their paradoxical lack of a $Na^+$/GTPγS shift (*Lee et al., 1999*). Finally, using the Black-Leff operational model (*Black and Leff, 1983*), the intrinsic efficacy (τ) was calculated using our published data from the [$^{35}$S] GTPγS assay (*Livingston and Traynor, 2014*). This variable can only be reliably calculated and separated from functional affinity ($K_A$) for partial agonists (*Figure 3d*). Using this limited dataset (*Figure 3—source data 1*), we observed a significant correlation ($r^2 = 0.83$) between τ and the $t_{1/2}$ of Nb39 association.

The above efficacy comparisons are all related to receptor-G-protein interactions. However, as mentioned above the µ-OR, like other GPCRs, may show a signaling bias in that ligands could preferentially activate G proteins or β-arrestin-mediated downstream signaling, this has been demonstrated for example with the newer ligands PZM21(*Manglik et al., 2016*) and TRV130 (*DeWire et al., 2013*). To determine if there was a correlation with β-arrestin recruitment, we compared published data of τ values for a series of eight opioid ligands (*McPherson et al., 2010*) with their Nb39 association rates from *Table 1*. For this dataset there was a significant correlation, albeit weaker ($r^2 = 0.62$, p=0.019; *Figure 3e*).

In addition to monitoring Nb39 association, the interferometry technique also allows for measurement of the dissociation of Nb39 from ligand-bound µ-OR (*Figures 1* and *2*). Compared to the wide range of association rates observed, the dissociation of Nb39 was generally constant across the various ligands, including full and partial agonists and compounds such as buprenorphine that have slower receptor dissociation (*Table 1*). As examples, the Nb39 dissociation rate ($t_{1/2}$) was the same

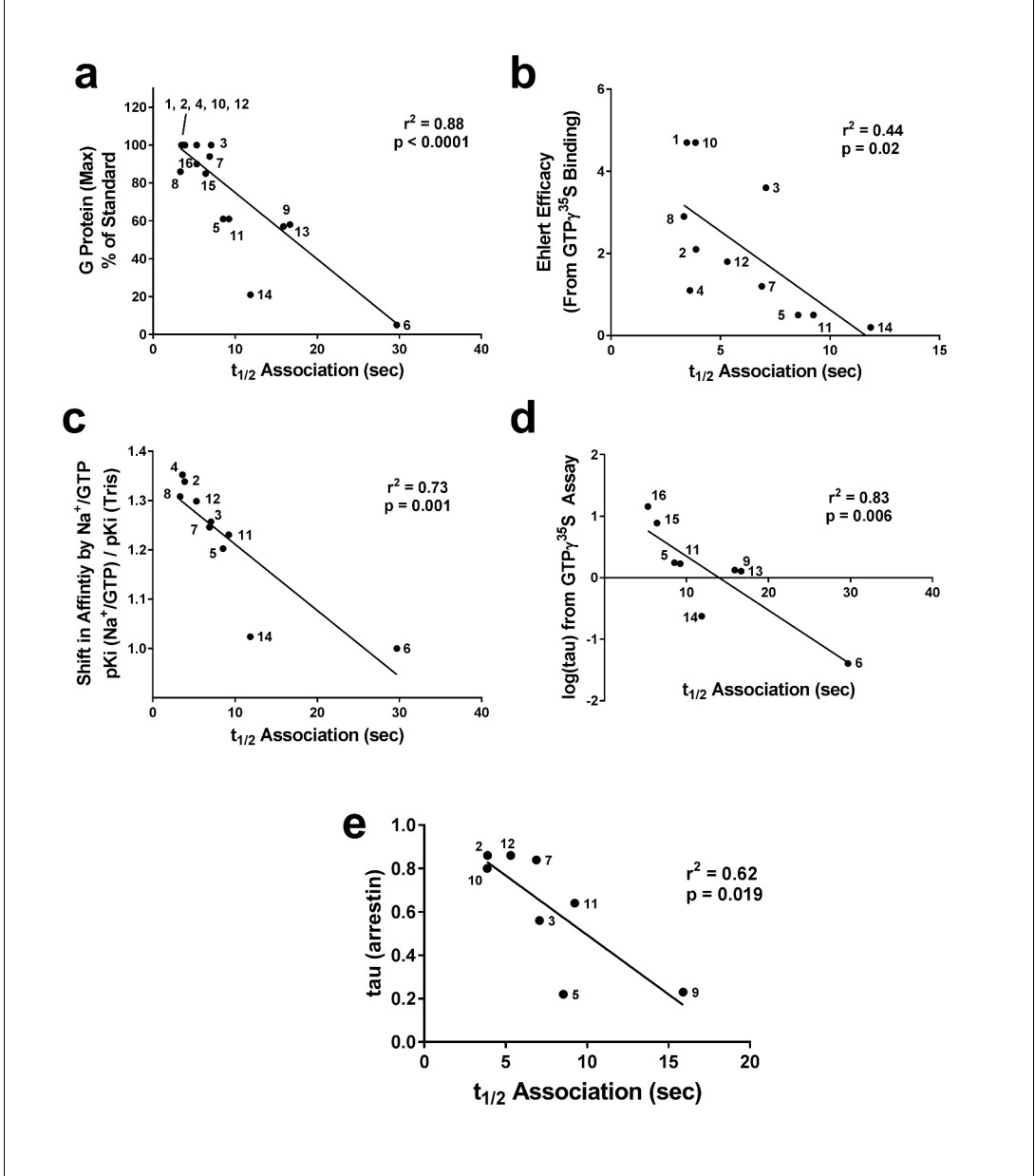

**Figure 3.** Correlation of association times of Nb39 with various measures of agonist efficacy. The $t_{1/2}$ of association of Nb39 in the presence of saturating agonist was measured and is plotted against (**a**) maximal stimulation of GTP$\gamma^{35}$S binding by agonist, (**b**) the calculated Ehlert efficacy (***Ehlert, 1985***) values for each agonist to activate G protein, (**c**) the shift in affinity of the agonist as measured by radioligand competition binding in the absence or presence of Na$^+$/Guanine nucleotide, and (**d**) the τ value using data from GTP$\gamma^{35}$S assays (***Livingston and Traynor, 2014***) or (**e**) β-arrestin recruitment assays; ***McPherson et al., 2010***), analyzed with the Black-Leff operational model (***Black and Leff, 1983***). Data used to compile correlation graphs is listed in the source data table. The ligands are: (1) BU72, (2) DAMGO, (3) Leu-Enk, (4) L-methadone, (5) Morphine, (6) Nalbuphine, (7) Endomorphin 2, (8) Loperamide, (9) Oxycodone, (10) Etorphine, (11) Fentanyl, (12) Met-Enk, (13) Hydrocodone, (14) Buprenorphine, (15) Morphine + BMS-986122, (16) Morphine + BMS-986187. Correlation analysis was performed using GraphPad Prism 6.02.

DOI: https://doi.org/10.7554/eLife.32499.005

The following source data is available for figure 3:

**Source data 1.** List of values used to construct correlation graphs.

DOI: https://doi.org/10.7554/eLife.32499.006

from both morphine-bound μ-OR (0.033 min$^{-1}$), the BU72-bound μ-OR (0.031 min$^{-1}$) and buprenorphine bound μ-OR (0.025 min$^{-1}$), despite the markedly different efficacies of these agonists. This may suggest that both full and partial agonists are capable of producing similar active states, although to different extents as indicated by the observed association rates, or that Nb39 is incapable of differentiating partial agonist from full agonist states. Unexpectedly, Nb39 dissociated more rapidly from L-methadone- and loperamide-bound μ-OR, with rates of 0.052 ± 0.003 min$^{-1}$ and 0.044 ± 0.003 min$^{-1}$ for L-methadone and loperamide, respectively (*Figure 2c*; *Table 1*). This difference can be interpreted as a distinct methadone-and loperamide-bound μ-OR conformation(s) with decreased affinity for Nb39 as compared to other ligands. To further explore this possibility, we utilized the biased opioid agonist PZM21(*Manglik et al., 2016*). The presence of PZM21 resulted in Nb39 recruitment and afforded dissociation kinetics (*Table 1*) similar to those of methadone, providing further evidence that different ligands may cause the receptor to interact with Nb39 in unique ways.

To ensure that $K_{obs}$ provides a viable surrogate for Nb39 interaction with μ-OR, we experimentally determined the true $K_{on}$, $K_{off}$, and $K_d$ of Nb39 for particular agonists (*Figure 4*). The agonists chosen included the highly efficacious BU72 and DAMGO, the partial agonist morphine, and methadone which showed atypical dissociation. The ligands displayed differences in their ability to produce binding of increasing concentrations of Nb39. In addition, the $K_{off}$ values given in the figure legend were similar to those shown in *Table 1*, with methadone as an outlier in both cases. The affinity of Nb39 for μ-OR varied for each ligand such that the calculated $K_{on}$ values followed the same pattern as $K_{obs}$, namely BU72 > DAMGO > methadone > morphine. From these data, it can be seen that $K_{obs}$ is a suitable parameter for efficacy determination.

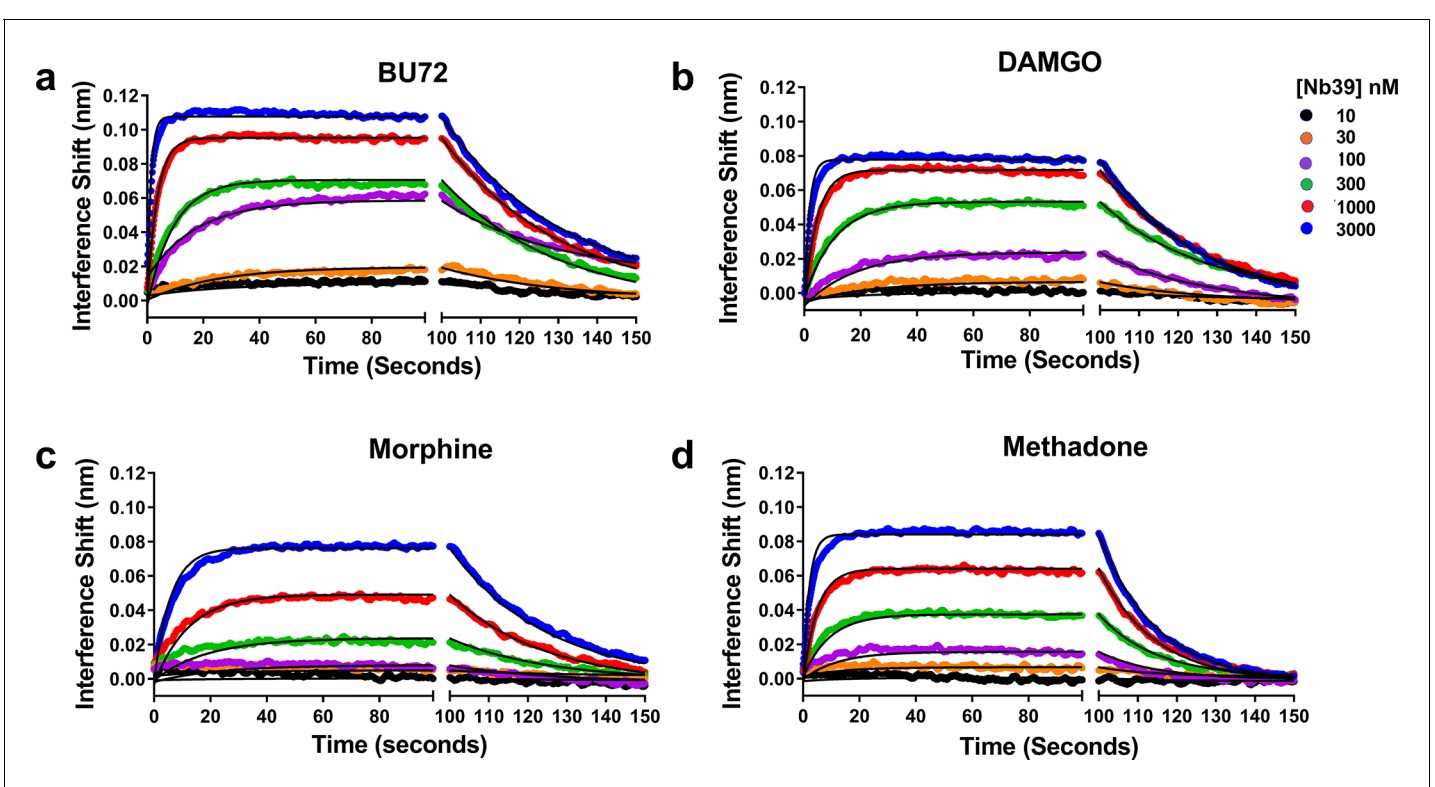

**Figure 4.** Association and dissociation of a range of Nb39 concentrations induced by various agonists. The association of six different concentrations of Nb39 was measured in the presence of saturating ligand concentrations. Utilizing global regression analysis, the $K_d$ of Nb39 (nM) for ligand bound receptor was calculated as follows: BU72 (144 ± 4), DAMGO (194 ± 9), Morphine (944 ± 13), Methadone (580 ± 3). The $K_{on}$ values (min$^{-1}$, M$^{-1}$ × 10$^{-5}$) were BU72 (2.35 ± 0.45), DAMGO (17.8 ± 0.5), Morphine (0.43 ± 0.05), and Methadone (1.21 ± 0.07) and the $K_{off}$ values (min$^{-1}$) BU72 (0.034 ± 0.0003), DAMGO (0.034 ± 0.003), morphine (0.04 ± 0.003), methadone (0.07 ± 0.0002).
DOI: https://doi.org/10.7554/eLife.32499.007

## Allosteric modulation of μ-OR in rHDL by small-molecule PAMs

Previously, we have suggested that the μ-PAM, BMS-986122, enhances the affinity and efficacy of orthosteric ligands by stabilizing active state(s) of μ-OR. To test this hypothesis using the binding of Nb39 to μ-OR, we first needed to validate that BMS-986122 had detectable allosteric activity at monomeric μ-OR in rHDL particles, as all previous data on the compound were generated using cell membrane preparations. The affinity of L-methadone for monomeric μ-OR determined from competition binding experiments using the opioid antagonist $^3$H-diprenorphine (DPN) in the presence or absence of 10 μM BMS-986122 was enhanced three-fold in the presence of 10 μM BMS-986122 (*Figure 5*). This shift is much smaller than seen in membranes prepared from C6 rat glioma cells stably expressing MOPr (*Burford et al., 2013*). In order to determine if this diminished BMS-986122 activity in the μ-OR-rHDL system was a property of BMS-986122 or a property of purified μ-OR, we investigated BMS-986187, another PAM that is structurally distinct from BMS-986122 (*Figure 5*). Although initially discovered as a PAM of the closely related delta opioid receptor (δ-OR), this compound is a weak PAM at the μ-OR (*Burford et al., 2015*). In contrast to BMS-986122, BMS-986187 produced a 12-fold enhancement of L-methadone affinity for μ-OR in the rHDL system (from a $K_i$ = 2696 (1970–3691) nM to 212 (142-317) nM), and shifted the affinity of DAMGO by 6-fold (from a $K_i$ of 1240 (632–2438) nM to 212 (142-317) nM), although it failed to alter the affinity of morphine, ($K_i$ = 630 nM in the absence and 500 nM in the presence of BMS-986197 (*Figure 5*). This probe dependence matches that seen with BMS-986122 in cell membranes, and can be most simply explained by a two-state model of GPCR function (*Monod et al., 1965*; *Livingston and Traynor, 2014*) in which the μ-PAMs stabilize R* states of μ-OR. We performed competition assays with L-methadone in the presence of increasing concentrations of BMS-986187 and applied the allosteric ternary complex model (GraphPad Prism) to calculate a cooperativity factor (α) of 58 and a $K_B$ of 4.5 μM, representing the affinity of BMS-986187 for the unoccupied μ-OR in rHDL. This is similar to data obtained for BMS-986187 at the μ-OR in cloned membranes ($K_B$ of 5.5 μM, log αβ of 1.16; *Livingston et al., 2018*) determined from a [$^{35}$S]GTPγS functional assay using a derivation of the allosteric ternary complex model (*Leach et al., 2010*). From these values, we determined the affinity of BMS-986187 for the methadone-bound μ-OR in rHDL ($K_B/\alpha$) to be 77 nM. This represents an increase of 58-fold in the affinity of the modulator in the presence of methadone, indicating a strong preference for the active R* state of the receptor.

We hypothesized that both PAMs stabilize active R* states of μ-OR but that BMS-986187 has an increased allosteric interaction with μ-opioid agonists compared to BMS-986122, as seen by the enhanced cooperativity with μ-opioid agonists in competition binding assays and in our previous functional and binding assays using cell membranes (*Livingston et al., 2018*). Therefore, we predicted that the allosteric ligands alone should promote the binding of Nb39, but that Nb39 binding in the presence of BMS-986187 would be more rapid and to a greater extent than in the presence of BMS-986122. Indeed, both allosteric ligands were able to cause Nb39 binding, although at a slower rate as compared to most orthosteric agonists (*Figure 6a*; *Table 1*). However, dissociation of Nb39 from the BMS-986122-bound or BMS-986187-bound μ-OR proceeded with a $k_{off}$ similar to Nb39 dissociation from μ-OR bound to orthosteric agonists (other than methadone, loperamide and PMZ21). This suggests that the active states of μ-OR stabilized by the PAMs may be similar to those stabilized by traditional orthosteric agonists, although different from those stabilized by L-methadone and PMZ21.

In addition to the ability of the PAMs to stabilize R* conformations of μ-OR alone, we were interested in investigating the cooperative effects of the allosteric ligands on the ability of orthosteric agonists to promote Nb39 binding. Since both allosteric ligands have the ability to increase the efficacy of various orthosteric ligands in cell-based signaling assays (*Livingston and Traynor, 2014*; *Burford et al., 2015*), we expected that this increase in efficacy would manifest as an increase in the observed association rate constant of Nb39, and that BMS-986187 would have a larger effect than BMS-986122. Shown in *Figure 6b* is the ability of the two μ-PAMs to enhance morphine-driven recruitment of Nb39. As predicted, both allosteric ligands enhanced the rate constant of Nb39 association. Using the association rates in the presence of modulator and data from cell-based GTPγ$^{35}$S-binding assays of morphine in the presence of the modulators, the τ values were plotted in *Figure 3e*. These agreed with the correlation, indicating the increase in association was accompanied by an increase in intrinsic efficacy.

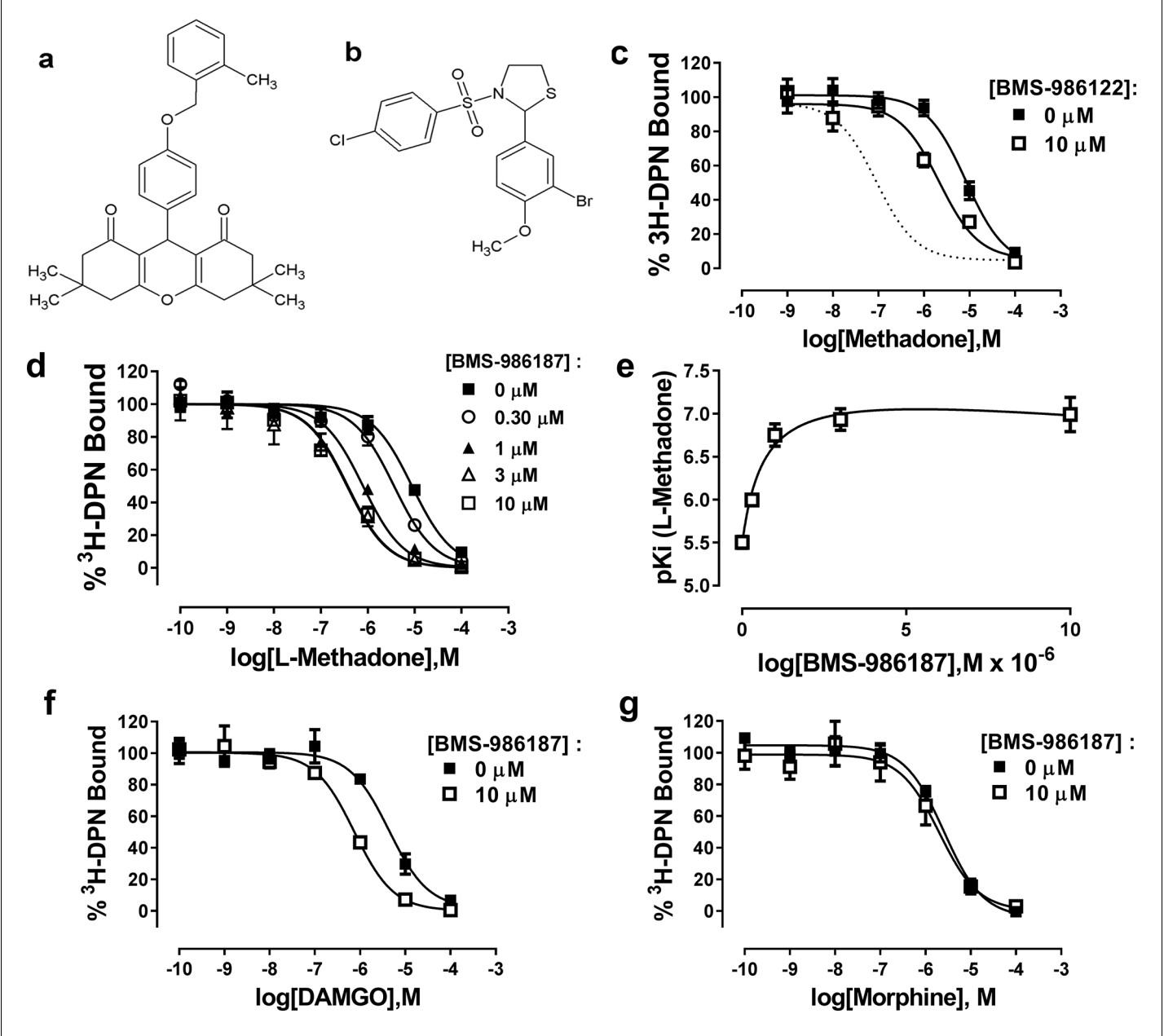

**Figure 5.** Allosteric modulation of μ-OR-rHDL by small molecule PAMs. Structures of (**a**) BMS-986187 and (**b**) BMS-986122. The ability of BMS-986122 (**c**) or BMS-986187 (**d**) to enhance the binding affinity of L-methadone was measured using displacement of the orthosteric antagonist [3]H-diprenorphine. The dotted line in (**c**) shows the effect previously obtained in membranes from C6μ cells (*Livingston and Traynor, 2014*). The effect of BMS-986187 on L-methadone affinity is plotted in (**e**). Enhancement of the affinity of DAMGO or morphine in the presence of 10 μM BMS-986122 is shown in (**f**) and (**g**), respectively. All plotted points are means ±s.e.m. of three (morphine) or five independent experiments (all other), each in duplicate. Nonlinear regression analysis with GraphPad Prism 6.02 was utilized to determine the affinity of the ligands. Hill slopes were not significantly different from unity.
DOI: https://doi.org/10.7554/eLife.32499.008

We repeated the same experiment with the orthosteric ligands L-methadone and DAMGO, but used a lower concentration of Nb39 (100 nM) to enhance sensitivity as these orthosteric ligands are of higher efficacy. Both PAMs enhanced the rate constant of DAMGO-driven Nb39 binding (*Table 2*) and slowed the dissociation of Nb39 from μ-OR, with BMS-986187 having a larger effect. Unexpectedly, the association of Nb39 to L-methadone-bound μ-OR was slowed in the presence of either

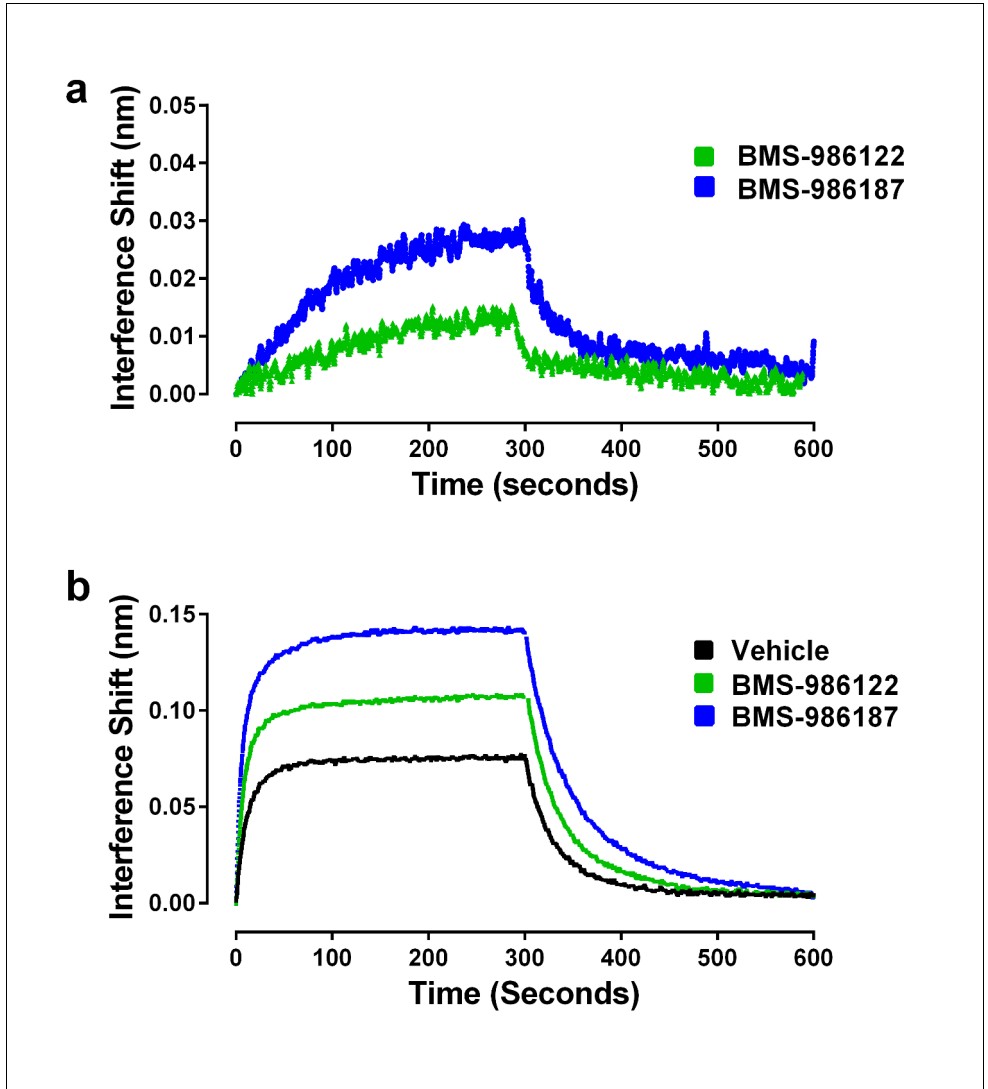

**Figure 6.** Effects of allosteric ligands on binding kinetics of Nb39. The association and dissociation of Nb39 (1 µM) was measured as described in the Materials and methods. (a) Shown is a representative experiment of data in *Table 1*, comparing two allosteric agonists at 30 µM. (b) Representative experiment comparing the kinetics of Nb39 binding in the presence of morphine in the absence (black), or presence of 30 µM BMS-986122 (green), or 30 µM BMS-986187 (blue).

DOI: https://doi.org/10.7554/eLife.32499.009

BMS-986122 or BMS-986187. Additionally, the dissociation of Nb39 was unchanged by BMS-986122 but slowed significantly by BMS-986187 (*Table 2*).

Finally, we sought to determine if we could use purified heterotrimeric G protein binding to µ-OR in place of the Nb39 biosensor. In order to allow for measures of association and dissociation, soluble heterotrimeric G protein composed of a complex of myristoylated Gαi1, β1, and the γ2 C68S mutant lacking prenylation was utilized (see Materials and methods). For this experiment, we studied BU72 and morphine as two ligands with differing efficacy. Each ligand was able to promote binding of 100 nM heterotrimeric G protein to µ-OR, though to different extents and there was low binding in the absence of ligand (*Figure 7a*). Utilizing a global fit of the data, the $K_d$ for G protein varied between ligands (see legend to *Figure 7*). In contrast to Nb39 which dissociated very rapidly, the dissociation $t_{1/2}$ of G protein was much slower in the presence of BU72 (231 min) or morphine (182 min). To test the hypothesis that this slow dissociation was due to formation of highly stable nucleotide-free receptor:G protein complexes (*Rasmussen et al., 2011b*), dissociation was measured in

**Table 2.** Alteration in Nb39 kinetics in the presence of µ-PAMs

**Morphine (1 µM Nb39)**

| | $k_{obs}$ (min$^{-1}$) | $t_{1/2}$Assoc (sec) | $k_{off}$ (min$^{-1}$) | $t_{1/2}$Diss(sec) | N |
|---|---|---|---|---|---|
| Vehicle | 0.08 ± 0.01 | 8.5 | 0.033 ± 0.001 | 21 | 3 |
| BMS-986122 | 0.11 ± 0.001 | 6.4 | 0.032 ± 0.0002 | 22 | 3 |
| BMS-986187 | 0.13 ± 0.01 *** | 5.3 | 0.024 ± 0.0004 | 29 | 3 |
| **L-Methadone (100 nM Nb39)** | | | | | |
| | $k_{obs}$ (min$^{-1}$) | $t_{1/2}$Assoc (sec) | $k_{off}$ (min$^{-1}$) | $t_{1/2}$Diss(sec) | n |
| Vehicle | 0.087 ± 0.009 | 8.0 | 0.050 ± 0.004 [†] | 14 | 7 |
| BMS-986122 | 0.077 ± 0.007 | 9.0 | 0.047 ± 0.004 +[+] | 15 | 7 |
| BMS-986187 | 0.055 ± 0.004 * | 13 | 0.033 ± 0.002 ** | 21 | 7 |
| **DAMGO (100 nM Nb39)** | | | | | |
| | $k_{obs}$ (min$^{-1}$) | $t_{1/2}$Assoc (sec) | $k_{off}$ (min$^{-1}$) | $t_{1/2}$Diss(sec) | n |
| Vehicle | 0.051 ± 0.003 | 14 | 0.035 ± 0.003 [‡] | 20 | 7 |
| BMS-986122 | 0.050 ± 0.01 | 13 | 0.029 ± 0.003 | 24 | 7 |
| BMS-986187 | 0.050 ± 0.004 | 14 | 0.022 ± 0.001* | 31 | 7 |

Values are means from independent experiments (number of individual experiments indicated in 'n' column). Analyses were performed by two-way ANOVA with a Tukey post-hoc test.

*Indicates significance compared to vehicle condition for each orthosteric ligand (*$p<0.05$, **$p<0.01$, ***$p<0.001$). +[+]Indicates $p<0.01$ as compared to L-methadone/BMS-986187 combination.

[†]Indicates $p<0.01$ as compared to morphine/vehicle combination.

[‡]Indicates $p<0.01$ as compared to DAMGO/vehicle combination.

DOI: https://doi.org/10.7554/eLife.32499.010

the presence of 1 µM GDP. The presence of the nucleotide rapidly enhanced G protein dissociation (*Figure 7b*).

## Discussion

We have described a method for examining the efficacy of orthosteric and allosteric ligands of a prototypic class A GPCR, µ-OR, which relies upon the ability of the conformationally-selective sensor Nb39 to recognize active (R*) state(s) of the receptor as confirmed using heterotrimeric G protein. Understanding the effects of ligands on the distribution of receptor conformations is crucial in predicting their activity at downstream signaling outputs and in vivo. Direct measurement of Nb39 binding to µ-OR using interferometry is independent of signaling and does not require calculations of efficacy from parameters determined in signaling assays. Further, the output of the interferometry assay is not subject to amplification and is dependent only upon receptor-ligand interaction, enabling detection of fine distinctions in agonist action that may be masked when measuring a downstream signaling output (*Black and Leff, 1983*; *Ehlert, 1985*). Importantly, the technique has the potential to detect differences in agonist-induced receptor conformations that may predict biased signaling without relying upon traditional calculations of 'bias factors' (*Kenakin et al., 2012*; *Stott et al., 2016*). Finally, the method can be applied to positive allosteric modulation of orthosteric ligands and provides insight into the mechanism of action of PAMs at µ-OR by showing these molecules, although acting at a distinct site, stabilize active (R*) receptor states even in the absence of orthosteric agonist. Overall, the method provides a way of quantitatively examining the efficacy of µ-OR orthosteric and allosteric ligands to stabilize R*, and can readily be applied to other GPCRs with conformationally-selective nanobodies available (*Staus et al., 2014*; *2016*).

As predicted based on data from previous studies with nanobodies (*Rasmussen et al., 2011a*; *Irannejad et al., 2013*; *Huang et al., 2015*), orthosteric agonists resulted in robust Nb39 binding, while the orthosteric antagonists naloxone or diprenorphine failed to promote detectable Nb39 binding. We show the ability of an agonist to promote Nb39 binding is correlated with its ability to promote signal transduction through $G_{i/o}$ protein as measured by GTPγ$^{35}$S binding. In fact,

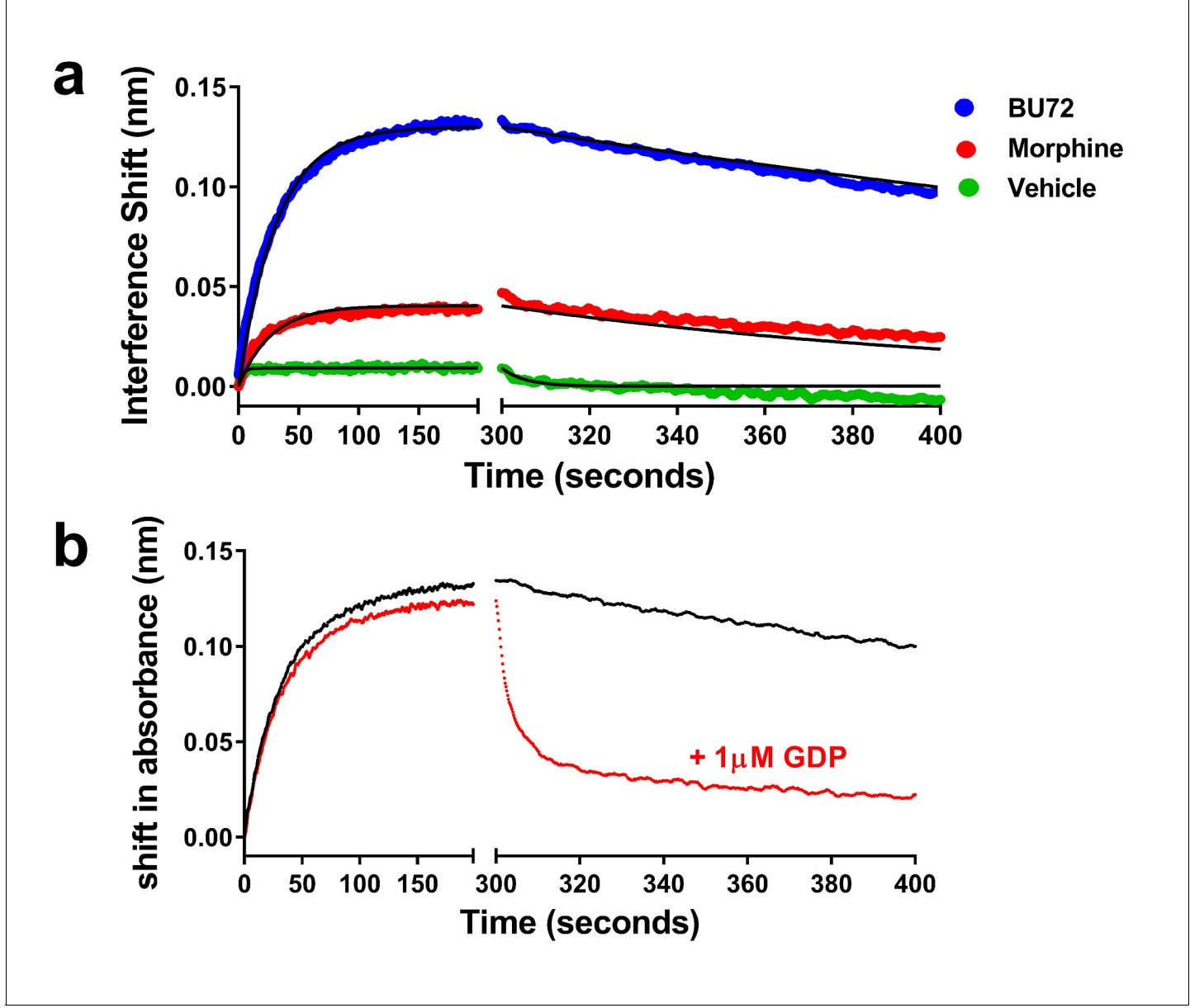

**Figure 7.** Association and dissociation of heterotrimeric G protein and the effect of GDP on dissociation. (**a**) Utilizing a global fit analysis the affinity ($K_d$) of G protein for μ-receptor was determined as 13 ± 1 nM for the BU72-bound receptor and 2.9 ± 0.8 nM for the morphine-bound receptor, though these are pseudo-affinity constants as the G protein binding was nearly irreversible on the time-scale studied. Only in the presence of nucleotide (**b**) did the BU72-bound heterotrimer rapidly dissociate.

DOI: https://doi.org/10.7554/eLife.32499.011

measurement of Nb39 binding is more sensitive than GTPγ[35]S binding, as some low-efficacy ligands, including nalbuphine and the μ-OR PAMs, were able to promote Nb39 binding but failed to show measurable G protein activity using the GTPγ[35]S-binding assay (*Livingston and Traynor, 2014*). While theoretically measuring the same process, that is μ-OR active-state promotion by agonists, leading to a protein-protein interaction at the receptor's cytoplasmic face, the output of the GTPγ[35]S assay can vary dramatically based upon nucleotide concentration, receptor:$Gi_{/o}$ protein ratios, $MgCl_2$ concentration, NaCl concentrations, time, and temperature (*Traynor and Nahorski, 1995*; *Szekeres and Traynor, 1997*; *Remmers et al., 2000*; *Heusler et al., 2016*). In comparison, Nb39 binding is not an enzymatic process like nucleotide exchange, but instead represents a

bimolecular binding event. However, it is recognized that measured efficacy in a cellular context will depend on the signal measured and the environment of the receptor.

It is important to note that the fact we see differences in $K_{obs}$ for the various agonists indicates that each agonist stabilizes a different distribution of active states of μ-OR, but that these states are all recognized by Nb39. The range of active μ-OR conformations formed by the orthosteric agonists and/or allosteric modulators recognized by Nb39 is, as of yet, unknown, and within a cellular context will depend on the environment and state of the receptor, but the crystal structure of active μ-OR bound to Nb39 shows that the nanobody does stabilize an active state of the receptor displaying the prototypic outward movement of TM6 associated with activation of GPCRs (*Farrens et al., 1996*; *Scheerer et al., 2008*; *Rasmussen et al., 2011b*; *Kruse et al., 2013*; *Huang et al., 2015*). Also, the affinities of the agonist BU72 for Nb39-bound μ-OR and Gαi-bound μ-OR are similar (*Huang et al., 2015*), suggesting the conformations of μ-OR stabilized by Nb39 and Gαi are also comparable, a finding confirmed by our current data with heterotrimeric G protein. The correlation with other efficacy measures leaves room for interpretation on the nature of receptor states that Nb39 can bind. If Nb39 was able to recognize and bind all active-like conformations capable of initiating downstream signal transduction, one might expect perfect correlations with other measures of efficacy. As evidenced, the correlation with Ehlert's efficacy values shows this is not the case. Although a well-accepted method for determining intrinsic efficacy, Ehlert's measure relies upon data collected from a signaling assay, therefore if a ligand has a signaling bias, the results may differ depending on which signaling output is chosen. The Ehlert equation relies upon a potency value ($EC_{50}$), affinity value, and an $E_{max}$ that is relative to a chosen standard. The $EC_{50}$ of an assay will be heavily reliant on the system: receptor reserve, time of incubation, and inherent system maximum and is therefore not an absolute number (*Strange, 2008*). Additionally, the affinity of the ligand is not a factor in the interferometry assay as all ligands are present at saturating concentrations. This is likely why the correlation with τ as calculated from the operational model is stronger (*Figure 4*). By calculating τ alone, ligand affinity is removed as a descriptor of efficacy.

Data from the interferometry experiments correlated strongly with the shift in affinity of orthosteric agonists in the presence of $Na^+$ ions and Guanine nucleotides. Affinity is ideally system-independent and so the $Na^+/GTPγS$ shift appears to be more reflective of true intrinsic efficacy, or ability of a ligand to discriminate between R and R*, with no dependence on the selection and measurement of a signaling output. It should be noted that two agonists, etorphine and BU72, are unique in that they exhibit little sensitivity to $Na^+/GTPγS$ despite their high efficacy. It is possible that their insensitivity to $Na^+$ ions is not absolute. Rather, etorphine and BU72 may be extremely negatively cooperative with $Na^+$ binding, thereby reducing the potency of $Na^+$ to alter the affinity of these ligands to such a degree that no shift was observed at the $Na^+$ concentration tested (100 mM). Nonetheless, the lack of the ability of $Na^+/GTPγS$ to alter affinity of etorphine and BU72 is in agreement with the lack of allosteric effects of the PAMs on these ligands (*Livingston and Traynor, 2014*), suggesting they drive formation of R* too efficiently to be further enhanced by allosteric ligands. Indeed, NMR data have suggested that BU72 is a 'superagonist' capable of stabilizing conformational changes in μ-OR, in particular increased dynamics in intracellular loop 1 and helix 8, that are not seen with other full agonists (*Sounier et al., 2015*). Etorphine may act in a similar fashion.

The rationale of the present study was to directly compare agonist ability to generate an active conformation of the μ-OR as a determinant of intrinsic efficacy without the need to measure a downstream signal. The comparative measures above are all indicative of Gα function. However, if measures are taken downstream that involve the Gβγ subunit then the actual value of the efficacy measure might vary but the rank order should stay the same since Nb39 mimics the action of heterotrimeric G proteins. It is possible that non-G-protein-mediated actions, that could include β-arrestin recruitment, would show a different rank order. Consequently, we also compared the kinetics of Nb39 binding induced by several agonists with published agonist efficacy data for β-arrestin recruitment (*McPherson et al., 2010*) and saw a significant correlation suggesting that Nb39 may recognize μ-OR states that recruit β-arrestin as well as those that recruit G protein, or that recruitment of β-arrestin to μ-OR must be preceded by transition to a Nb39-recognizable state. We did not necessarily expect this finding given that nanobodies which promote high affinity agonist binding of the β2AR recognize a conformation of the receptor similar to the conformation which binds G protein (*Rasmussen et al., 2011a*). On the other hand the result does support the finding that some ligands, including the biased agonist PZM21, show different kinetics. It will be important in the future to

develop the system to follow β-arrestin recruitment to purified μ-OR directly. However, this will require the purification of homogenous, phosphorylated receptor.

Like the orthosteric agonists, the allosteric ligands BMS-986122 and BMS-986187 also drove Nb39 binding to μ-OR suggesting they can directly activate the receptor to form R*, even in the absence of an agonist occupying the orthosteric site. However, the rate and extent of Nb39 binding produced by the PAMs was low, comparable to low-efficacy orthosteric agonists such as nalbuphine. Compared to BMS-986122, BMS-986187 promoted Nb39 binding to a greater extent and at a faster rate, in line with its higher activity to enhance orthosteric ligand binding affinity to μ-OR in rHDL particles. The fact that BMS986187 is able to recruit Nb39, albeit to a much smaller extent than orthosteric agonists, identifies this compound as an ago-PAM, a compound that can activate receptor in the absence of orthosteric agonist and demonstrates the high sensitivity of the assay. This is supported by our recent observation that BMS986187 acts as an agonist in the adenylate cyclase assay but not in an assay for G-protein activation of β-arrestin recruitment (*Livingston et al., 2018*). Our proposed mechanism for PAMs at the μ-OR is to allosterically modulate the $Na^+$ ion binding site and therefore drive or help to drive a active conformational (R*) ensembles. Thus, there is a fine line between a PAM and an ago-PAM that predicts a continuum between compounds that are silent (SAMs) and bind to the allosteric site without any obvious activity, PAMs that promote agonist action and ago-PAMs. Moreover, these definitions will depend on the efficacy requirements of the system.

When added together with orthosteric ligands the PAM, BMS-986187, increased the association rate of Nb39 to μ-OR induced by morphine, but did not alter the rate of Nb39 binding in the presence of DAMGO and decreased the effect of association rate in the presence of methadone. In previous studies, we showed the efficacy of the partial agonist morphine, determined using Ehlert's equation, was increased two-fold by BMS-986122 (from e = 0.5 to e = 1.0), whereas the efficacy of DAMGO (e = 2.0 versus 2.1) was essentially unchanged in the presence of BMS-986122 and L-methadone was slightly decreased (from e = 1.1 to 0.9). Instead with L-methadone and DAMGO there is an increase in potency. Crystallographic work is underway to determine the mode of PAM binding and to determine the structural features that govern allosteric modulation of μ-OR.

In addition to association rate constants, valuable information can be obtained from the rates of Nb39 dissociation since we can assume this rate is a product of the affinity of Nb39 for agonist-bound and active μ-OR. The dissociation rate of Nb39 was equal in the presence of most orthosteric and the allosteric ligands examined in this study, arguing that Nb39 recognizes all active (R*) conformations equally and/or that both full and partial orthosteric agonists, as well as the allosteric modulators, stabilize a common ensemble of μ-OR active conformations that is recognized by Nb39. These arguments are consistent with the exception of L-methadone, loperamide and PZM21 which show faster Nb39 dissociation. This indicates that the receptor has decreased affinity for Nb39 in the presence of these ligands, possibly reflecting a unique set of receptor conformations. In support of a differential binding mode for L-methadone, this ligand is the most sensitive orthosteric agonist toward allosteric modulation with three different chemical scaffolds (BMS-986122, BMS-986187, and MS1 [*Bisignano et al., 2015*]), and in turn greatly enhances the affinity of BMS-986187 for the allosteric site. This suggests that L-methadone, and possibly PZM21 and loperamide, could engage with μ-OR in distinct ways, generating unique conformational ensembles that can be seen in its effects on the binding characteristics of both Nb39 and small-molecule allosteric modulators. It is possible that other factors, including differential rates of ligand dissociation could be confounding factors in Nb39 dissociation, but the fact that these differences are only seen for three ligands would argue against this. Although we have tested a number of structurally diverse μ-OR orthosteric ligands, and both full and partial agonists, it is feasible that each ligand generates distinct active conformations that are indistinguishable to Nb39, including as discussed above the possibility of β-arrestin preferring conformations. This is likely given evidence at the β2-adrenergic (*Yao et al., 2006*) and β1-adrenergic receptors (*Warne et al., 2011*) suggests partial agonist conformations differ from full agonist-bound conformations. Thus, it should be possible to develop selective nanobodies that recognize agonist-specific μ-OR* states. If true it is feasible that these different conformational states may contribute to the lower levels of cross-tolerance observed with methadone compared to other opioids in animal models (*Neil, 1982*; *Posa et al., 2016*), and the higher relative potency values for methadone than expected during opioid rotation in patients (*Knotkova et al., 2009*). Moreover, these different receptor states could explain the reported biased agonist nature of PMZ21(*Manglik et al., 2016*). However, this represents the most basic piece of the puzzle and in the whole cell many

factors not present in rHDL particles, such as differential posttranslational processing of the receptor, and the presence of accessory proteins, as well as different membrane lipids could contribute to the ensemble of receptor conformations obtained.

The affinity of Nb39 for μ-OR (*Huang et al., 2015*) is low compared to Nb80 at the β2AR (*Manglik et al., 2016*). Thus, Nb39 does not bind receptor in the absence of agonist and so does not create agonist conformations of the receptor or force orthosteric agonist to bind to these unique conformations. Although it is possible that the agonist bound Nb39 conformation in the presence of Nb 39 could be different from the agonist and heterotrimerc G-protein-bound conformations, our findings using a mutated, soluble form of heterotrimeric G protein confirmed the biological relevance of our model system by demonstrating that Nb39 is a suitable surrogate for heterotrimeric G proteins which are the native 'active-state sensors' of R* states of GPCRs. However, for these assays protein tools such as nanobodies are preferred due to the challenges associated with using G-protein complexes made up of three subunits. These include selection of subunits, which themselves could cause differences in receptor binding (*Remmers et al., 2000*), inclusion or exclusion of nucleotide, selection and concentration of nucleotide, and nonspecific binding. Indeed, some native Gα subtypes and wild-type Gγ cannot be used as obligatory lipid groups (palmitoylation, geranyl-geranylation) require detergents which would disrupt the rHDL particles, and also could drive direct lipid-lipid association with the rHDL particles. This creates the possibility for multiphase association and dissociation kinetics which can make data analysis difficult, for example the $K_d$ value may be an amalgamation of two distinct values derived from binding of G protein to receptor and to the rHDL discs. Moreover, heterotrimeric G-protein binding is not simply a bimolecular interaction between G protein and receptor but an enzymatic process in which nucleotide can dramatically influence rates and extents of binding. The extremely slow dissociation of heterotrimeric G protein makes the $K_d$ calculated a 'pseudo' rate constant as the reaction is irreversible on the time scale of the assay in the absence of nucleotide.

In summary, we have described a novel method for the quantitative evaluation of efficacy of both orthosteric and allosteric ligands using purified μ-OR in rHDL particles. This biophysical technique is also able to identify ligands, in particular L-methadone and PZM21, that induce distinct conformations of μ-OR. Allosteric modulators of μ-OR are themselves capable of generating active conformations of μ-OR competent to bind Nb39 and of driving G-protein-independent high-affinity agonist binding. This technique is more sensitive than traditional measures of efficacy and is not reliant upon signal amplification. The methodology should be applicable to a wide variety of GPCRs.

# Materials and methods

**Key resources table**

| Reagent type (species) or resource | Designation | Source or reference | Identifiers | Additional information |
|---|---|---|---|---|
| Peptide, recombinant protein | mu-opioid receptor | *Manglik et al. (2012)* | OPRM1 | |
| Peptide, recombinant protein | Gbetagamma | *Iñiguez-Lluhi et al., 1992* | | |
| Peptide, recombinant protein | Myristoylated Gα$_{i1}$ | *Greentree and Linder, 2004* | | |
| Peptide, recombinant protein | Apolipoprotein-AI | *Kuszak et al., 2009* | | |
| Peptide, recombinant protein | [D-Ala2, N-Me-Phe4, Gly5-ol]-Enkephalin acetate salt (DAMGO) | Sigma | E7384 | CAS#100929-53-1 |
| Peptide, recombinant protein | Leu-Enkephalin | Sigma | L9133 | |
| Peptide, recombinant protein | Met-Enkephalin | Sigma | M6638 | CAS#82362-17-2 |

*Continued on next page*

*Continued*

| Reagent type (species) or resource | Designation | Source or reference | Identifiers | Additional information |
|---|---|---|---|---|
| Peptide, recombinant protein | Endomorphin 2 | Sigma | SCP0133 | |
| Peptide, recombinant protein | Nanobody 39 (Nb39) | *Huang et al., 2015* | | |
| Chemical compound, drug | [3H]-diprenorphine | Perkin Elmer | NET1121250UC | |
| Chemical compound, drug | Morphine sulfate | National Institute on Drug Abuse, NIH. Drug Supply Catalog | 9300–001 | CAS # 6211-15-0 |
| Chemical compound, drug | (R)-Methadone | National Institute on Drug Abuse, NIH. Drug Supply Catalog | 9250–005 | CAS# 125-58-6 |
| Chemical compound, drug | Buprenorphine | National Institute on Drug Abuse, NIH. Drug Supply Catalog | 9064–110 | CAS# 53152-21-9 |
| Chemical compound, drug | BU72 | *Huang et al. (2015)* | | |
| Chemical compound, drug | Diprenorphine | Other | | CAS# 14357-78-9: Opioid Research Center, U Michigan |
| Chemical compound, drug | Etorphine | Other | | CAS# 14521-96-1: Opioid Research Center, U Michigan |
| Chemical compound, drug | Fentanyl | National Institute on Drug Abuse, NIH. Drug Supply Catalog | 9801–001 | CAS# 1443-54-5 |
| Chemical compound, drug | Hydrocodone | Other | | CAS# 125-29-1:Opioid Research Center, U Michigan |
| Chemical compound, drug | Loperamide | Other | | CAS # 34552-83-5: Opioid Research Center, U Michigan |
| Chemical compound, drug | Nalbuphine | Other | | CAS# 23277-43-2: Opioid Research Center, U Michigan |
| Chemical compound, drug | Naloxone | Sigma | PHR1802 | CAS# 51481-60-8 |
| Chemical compound, drug | Oxycodone | Other | | CAS# 76-42-6: Opioid Research Center, U Michigan |
| Chemical compound, drug | PZM21 | *Manglik et al. (2016)* | | |
| Chemical compound, drug | BMS-986187 | Bristol Myers Squib; | *Burford et al. (2015)* | CAS# 684238-37-7 |
| Chemical compound, drug | BMS-986122 | Bristol Myers Squib; | *Burford et al. (2013)* | CAS# 313669-88-4 |
| Software, algorithm | GraphPad Prism 6.0 | GraphPad, La Jolla, CA | https://www.graphpad.com/scientific-software/prism/ | |
| Software, algorithm | Octet Data Analysis 7.0 software | Pall Forte Bio | https://shop.fortebio.com/site-license-octet-data-analysis-software-version-7.x.html | |

## Materials

[3H]-Diprenorphine (DPN) was from PerkinElmer Life Sciences. BMS-986122 and BMS-986187 (structures in *Figure 5*) were synthesized or obtained as previously described (*Burford et al., 2013*;

2015). Fentanyl, morphine, methadone and buprenorphine were from the NIDA Drug supply; other opiates were from the Opioid Basic Research Center at the University of Michigan. All other chemicals, unless otherwise specified, were purchased from Sigma (St. Louis, MO).

## Purification of μ-OR

Full-length *Mus musculus* μ-OR bearing an amino-terminal FLAG epitope tag and a carboxy-terminal 6xHis tag was expressed in Sf9 insect cells (Invitrogen) using the Best Bacbaculovirus system (Expression Systems). A tobacco etch virus (TEV) protease recognition sequence was inserted after residue 51 and a rhinovirus 3C protease recognition sequence was inserted before residue 359 for cleavage during purification. Insect cells were infected with baculovirus encoding μ-OR 48–60 hr at 27°C. Receptor was solubilized and purified in a final buffer comprised of 25 mM HEPES pH 7.4, 100 mM NaCl, 0.01% MNG (Anatrace), and 0.001% cholesterol hemisuccinate (CHS), as previously described (*Manglik et al., 2012*).

## Purification of Nb39

Nb39 was purified as described (*Huang et al., 2015*). Briefly, Nb39 bearing a carboxy-terminal His tag was expressed in the periplasm of *Escherichia coli* strain WK6 grown in Terrific Broth medium containing 0.1% glucose, 2 mM $MgCl_2$, and 50 mg/ml ampicillin and induced with 0.5 mM isopropyl-b-D-thiogalactoside (IPTG). Cells were harvested after overnight growth at 25°C and incubated in a buffer containing 200 mM Tris, pH 8.0, 0.5 mM EDTA, 500 mM sucrose for one h on ice. Bacteria were osmotically lysed by rapid dilution in water. The periplasmic fraction was isolated by centrifugation of cell debris and was supplemented with NaCl (300 mM final) and imidazole (10 mM final). Nb39 was isolated from the periplasmic fraction by nickel affinity chromatography, and subsequently purified by size-exclusion chromatography in a buffer comprised of 20 mM HEPES pH7.5 and 100 mM NaCl. Peak fractions were pooled and concentrated to approximately 1 mM.

## Apolipoprotein purification and biotinylation

Apolipoprotein-AI (Apo-AI) was purified as described previously (*Kuszak et al., 2009*). Apo-A1 was biotinylated using NHS-PEG4-biotin (Pierce Biotechnology) at a 1:1 molar ratio. Following a 30 min biotinylation reaction at room temperature, the sample was dialyzed to remove free biotin.

## Purification and formation of G-protein heterotrimeric complex

Myristoylated $G\alpha_{i1}$ containing a hexahistidine tag inserted at residue 121 (*Kozasa and Gilman, 1995*) was expressed in *Escherichia coli* and purified as described (*Greentree and Linder, 2004*). To prepare Gβγ subunit lacking the geranyl-geranyl modification, *Trichoplusiani* cells (High Five; Invitrogen) were infected with baculovirus encoding for $G\beta_1$ and $His_6$-$G\gamma_2$C68S (*Iñiguez-Lluhi et al., 1992*) at an MOI of 1 for each virus. Cells were harvested ~48 hr post-infection and lysed by nitrogen cavitation in a buffer containing 50 mM HEPES (pH 8.0), 65 mM NaCl, 5 mMβ-mercaptoethanol (β-ME), and protease inhibitors (35 μg/ml phenylmethylsulfonyl fluoride, 32 μg/ml each N-tosyl-L-phenylalanine chloromethyl ketone and N-tosyl-L-lysine chloromethyl ketone, 3.2 μg/ml each leupeptin and soybean trypsin inhibitor). The lysate was centrifuged for 10 min at 1000 *g* and the resulting supernatant was centrifuged for 40 min at 100,000 *g*. The clarified lysate was supplemented with NaCl to a final concentration of 300 mM and applied to a packed column of cobalt-NTA resin (TALON;Clontech) pre-equilibrated with wash buffer containing 20 mM HEPES (pH 8.0), 300 mM NaCl, 5 mM β-ME, and protease inhibitors. The column was washed with 10 column volumes of wash buffer then eluted with a buffer composed of 20 mM HEPES (pH 8.0), 50 mM NaCl, 150 mM imidazole, 5 mM β-ME, and protease inhibitors. Fractions were analyzed by SDS-PAGE, and those containing Gβγ were pooled and diluted to a final volume of 50 ml using 20 mM HEPES (pH 8.0), 5 mM β-ME. The diluted fractions were loaded onto a MonoQ HR 10/10 (GE Life Sciences) pre-equilibrated with 20 mM HEPES (pH 8.0) and eluted using a linear gradient of NaCl in the same buffer. Fractions containing Gβγ were pooled, concentrated using an Amicon Ultra centrifugal filter, and applied to a Superdex S200 XK 16/70 (GE Life Sciences). Size exclusion chromatography was performed using a buffer composed of 20 mM HEPES, 100 mM NaCl, 1 mM EDTA, and 100 μM Tris(2-carboxyethyl)phosphine (TCEP). Peak fractions were pooled and concentrated to ~5 mg/ml as determined by Bradford assay. Concentrated protein was flash-frozen in liquid nitrogen and stored at −80°C until use. Complexes

of myr-G$\alpha_{i1}$ and G$\beta\gamma$ were prepared by mixing the subunits at a 1.2:1 molar ratio in a buffer containing 20 mM HEPES (pH 8.0), 100 mM NaCl, 1 mM EDTA, 1.1 mM MgCl$_2$, 10 μM GDP, and 100 μM TCEP. Following a 30 min incubation at 4°C, complexes were isolated by size exclusion chromatography using a Superdex S200 HR 10/30 (GE Life Sciences). Peak fractions were pooled, concentrated using an Amicon Ultra centrifugal filter, and flash frozen in liquid nitrogen for storage at −80°C.

## μ-OR-rHDL Reconstitution

Purified μ-OR was reconstituted into high-density lipoprotein (HDL) particles using biotinylated Apo-AI and the lipids 1-palmitoyl-2-oleoyl-sn-glycero-3-phosphocholine (POPC) and 1-palmitoyl-2-oleoyl-sn-glycero-3-phosphoglycerol (POPG) (both from Avanti Polar Lipids) in a 3:2 molar ratio as previously described (Whorton et al., 2007). For OctetRed® experiments, rHDL particles containing receptor were separated from empty rHDL by anti-FLAG affinity chromotography and elution fractions positive for $^3$H-diprenorphine binding were pooled.

## Nb39 kinetic assays

Nb39 binding to μ-OR in the presence or orthosteric and/or allosteric ligands was measured using the Octet RED biolayer interferometry system (Pall ForteBio). In this assay, μ-OR-containing biotinylated rHDL particles are immobilized on a streptavidin-coated fiber optic probe that is incubated into buffers containing ligands in the presence or absence of Nb39. Different reconstituted receptor preparations were tested with each ligand to ensure the kinetics did not vary based on the preparation. Dissociation of bound Nb39 was initiated by placing the probe in buffer containing ligands but no Nb39. Specifically, biosensors (Pall ForteBio) were loaded with biotinylated μ-OR-rHDL particles for 15 min at room temperature and the biosensors were transferred to the Octet RED instrument. Sensors were placed into assay buffer (20 mM HEPES, pH 7.7, 100 mM NaCl, 1 mM EDTA, 0.05% (w/v) BSA) with vehicle or various orthosteric/allosteric ligands for 10 min to reach equilibrium, unless stated otherwise. To measure Nb39 association, the probe was transferred to assay buffer with Nb39 (at indicated concentrations) for 5 min, followed by a 10 min dissociation step in assay buffer (preliminary studies showed that dissociation of Nb39 was quite rapid). All ligands (orthosteric and allosteric), once introduced to the probe, remained in each subsequent buffer during association and dissociation. All experiments were carried out at 25°C with the assay plate shaking at 2000 r.p.m. Non-specific binding was measured using a vehicle control with no ligands and this was subtracted to account for baseline drift. Raw data were processed to remove baseline using Octet Data Analysis 7.0 software (Pall Forte Bio) and exported to GraphPad Prism 6.0 for curve fitting of association and dissociation using a global linear regression analysis of the families of curves. The number of independent experiments is listed in the figure legends or tables. No statistical methods were used to predetermine sample size.

## Radioligand binding assays

For competition binding experiments in μ-OR- rHDL, a mixture of μ-OR-rHDL and $^3$H-diprenorphine ($^3$H-DPN) was incubated with varying concentrations of agonist in a binding buffer comprised of 25 mM HEPES pH 7.4, 100 mM NaCl, and 0.1% BSA in the presence or absence of 3 μM Nb39. For assays performed using cell membranes, conditions listed were kept the same except for exclusion of BSA and inclusion of 10 μg protein per well. Binding reactions were incubated for 2 h at 25°C. Free radioligand was separated from bound radioligand by rapid filtration onto a Whatman GF/C filter pretreated with 0.1% polyethylenimine using a 24-well harvester (Brandel). Nonspecific binding was measured in the presence of 10 μM naloxone, an opioid antagonist. Radioligand activity was measured by liquid scintillation counting using a Wallac 1450 MicroBeta counter (Perkin Elmer). A minimum of three independent experiments, each in duplicate, were performed and the values were pooled to generate the mean curves displayed in the figures. Competition binding data were fit to a one-site model using GraphPad Prism 6.0. Data are presented as means with 95% confidence limits in parentheses. No statistical methods were used to predetermine sample size.

## Acknowledgements

This work was supported by the National Institutes of Health grant R37 DA039997 (to JRT) and an EDGE award (to KEL) from the Endowment for the Basic Sciences, University of Michigan. KEL was

supported by the National Institutes of Health Training grant T32 DA007267, JPM was supported by T32GM007767 and the AHA Midwest Affiliate Predoctoral Fellowship (13PRE17110027). We thank Drs. Neil Burford and Andrew Alt at Bristol-Myers Squibb for the gifts of BMS-986122 and BMS-986187. We thank Dr. John Tesmer at University of Michigan for use of equipment and facilities and Dr. Jorge Iñiguez-Lluhí for assistance in kinetic analysis.

## Additional information

### Funding

| Funder | Grant reference number | Author |
|--------|------------------------|--------|
| National Institutes of Health | T32 DA007267 | Kathryn E Livingston |
| American Heart Association | 13PRE17110027 | Jacob P Mahoney |
| National Institutes of Health | T32GM007767 | Jacob P Mahoney |
| National Institutes of Health | R37 DA039997 | John R Traynor |

The funders had no role in study design, data collection and interpretation, or the decision to submit the work for publication.

### Author contributions

Kathryn E Livingston, Conceptualization, Data curation, Formal analysis, Methodology, Writing—original draft, Writing—review and editing; Jacob P Mahoney, Conceptualization, Writing—review and editing; Aashish Manglik, Resources, Visualization, Writing—review and editing; Roger K Sunahara, Resources; John R Traynor, Conceptualization, Supervision, Funding acquisition, Methodology, Writing—original draft, Writing—review and editing

### Author ORCIDs

Kathryn E Livingston (iD) http://orcid.org/0000-0002-8652-4045
John R Traynor (iD) http://orcid.org/0000-0002-1849-8316

### Decision letter and Author response

Decision letter https://doi.org/10.7554/eLife.32499.014
Author response https://doi.org/10.7554/eLife.32499.015

## Additional files

### Supplementary files

• Transparent reporting form
DOI: https://doi.org/10.7554/eLife.32499.012

### Data availability

All data generated and analyzed during the study are included in the manuscript and supporting files. Source files have been provided for Fig 3.

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
