## [Decision Letter]

Thank you for submitting your article "Measuring Ligand Efficacy at the Mu-Opioid Receptor Using a Conformational Biosensor" for consideration by *eLife*. Your article has been favorably evaluated by Philip Cole (Senior Editor) and three reviewers, one of whom, Volker Dötsch (Reviewer #1), is a member of our Board of Reviewing Editors. The following individuals involved in review of your submission have agreed to reveal their identity: Christopher Evans (Reviewer #2); Terry Kenakin (Reviewer #3).

The reviewers have discussed the reviews with one another and the Reviewing Editor has drafted this decision to help you prepare a revised submission.

Overall, the reviewers liked your approach and consider it an important advance in the field. Some questions, however, remain. In particular the dose response mentioned in question 1 should be carefully addressed.

Summary:

The paper describes a methodology for assessing ligand-induced activated conformation(s) of the Mu-opioid receptor using the binding of a nanobody previously described for stabilizing the active conformation of the isolated receptor for crystal analysis. This is an interesting and informative approach to measure discrete aspects of ligand interactions. The approach can assess characteristics such as ability to determine association and dissociation kinetics of "active states" that are likely to be important for ligand characteristics as a result of different complexes formed and post-translational modifications.

Essential revisions:

1) What are the effects of agonist concentration above and below the saturating concentrations that were used? This may be especially important for partial agonists or those with slow kinetics. Some agonist dose-response for nanobody association and dissociation should be presented for drugs comparing high or low affinity agonists and high or low efficacy agonists to determine how the assay responds to the dose variable.

2) The cellular milieu and complex formation of the receptor with other proteins including kinases, arrestins and calmodulins is likely to alter a receptors ability to assume agonist conformations as are the extent of different post-translational modifications. Thus, like the crystal structures, the assay proposed is lacking many of the interactions that could regulate agonist signaling and this assay is undertaken in a privileged and non-environmental working condition of the receptor. Similarly, the nano body can also bind to the GPCR by conformational remodeling of the conformational ensemble that a specific ligand is inducing. If this is the case the binding affinity and association rates would be biased by this remodeling process. These issues should be discussed by the authors in the manuscript.

3) Since µ-PAMs alone promote Nb39 binding but show no signaling is it accurate to define this assay as one measuring efficacy/intrinsic activity?

4) The evidence for ligand-induced unique/distinct conformations is not clear. There are kinetics of ligand association/dissociation, ion interactions, as well as the nanobody associations and dissociations which are the measured output. Without knowing the kinetics of all players how can alternative conformations be assumed as opposed to duration spent in common conformational states?

5) In many PAM programs, some of the members are PAM Agonists, i.e. it is a fine line between making it easier for an agonist to promote an active state vs. actually having the PAM itself induce the active state. It would be nice if the authors expanded on their thoughts on this a little more.

6) It is not clear exactly which 'allosteric binding model' is used in this study - standard Graphpad model? Is this the Hall model or more comprehensive model? This should be specified. Also, the authors discuss 'allosteric efficacy' and this is confusing since this implies beta effects from the allosteric functional model whereas it appears only alpha effects where measured – alpha is only part of the story as beta effects can increase or decrease ternary and quaternary complex production. In this regard, the authors need to specify why only affinity measurements should reflect everything there might be to know about the quality of the active state ensemble.

7) In general, more discussion of when only G protein alpha effects reflect total opioid agonist efficacy would make this a stronger paper.

---

## [Author Response]

Essential revisions:1) What are the effects of agonist concentration above and below the saturating concentrations that were used? This may be especially important for partial agonists or those with slow kinetics. Some agonist dose-response for nanobody association and dissociation should be presented for drugs comparing high or low affinity agonists and high or low efficacy agonists to determine how the assay responds to the dose variable.

The aim of this study was to determine if we could use nanobody as a mimic for G protein and so measure the rate of nanobody binding as an indicator of agonist efficacy. For the purposes of this present manuscript we used a maximal concentration of agonist to ensure rapid agonist binding and high agonist occupancy of the receptor (this is outlined on in the third paragraph of the subsection “Measure of orthosteric agonist efficacy using an interferometry-based technique”) and to obtain the maximal kinetics of Nb39 association to the agonist bound state of the receptor and its dissociation from the receptor. Also partial agonists such as nalbuphine and the very slowly dissociating partial agonist buprenorphine show the same off rates of Nb39 binding as the other (full) agonists (Table 1) suggesting that agonist binding kinetics are not a confounding issue. This is highlighted in the seventh paragraph of the aforementioned subsection. Use of lower concentrations to obtain dose-response relationships will make it difficult to separate the kinetics of ligand binding from the kinetics of Nb39 binding, and will result in changes in fractional occupancy of the agonist (according to each agonist’s cooperativity with Nb39) over the time course of our measurement. However, we agree such experiments could be informative and we are planning to tackle these in the future, but they will provide different information to that we were seeking in this present study.

2) The cellular milieu and complex formation of the receptor with other proteins including kinases, arrestins and calmodulins is likely to alter a receptors ability to assume agonist conformations as are the extent of different post-translational modifications. Thus, like the crystal structures, the assay proposed is lacking many of the interactions that could regulate agonist signaling and this assay is undertaken in a privileged and non-environmental working condition of the receptor. Similarly, the nano body can also bind to the GPCR by conformational remodeling of the conformational ensemble that a specific ligand is inducing. If this is the case the binding affinity and association rates would be biased by this remodeling process. These issues should be discussed by the authors in the manuscript.

We agree that the environment of the receptor and any modifications to this or accessory proteins will modulate the ability of receptor to assume agonist conformations and these may therefore be different from the purified protein situation. It would be impossible to mimic every environment that could occur in different μ-OR containing neurons across the nervous system. We have therefore added a caveat to our observations on receptor conformations in the eighth paragraph of the Discussion.

However, this diversity in cellular environments and the impact of environment on signaling parameters is something we are trying to avoid by using a reductionist approach as stated in the second paragraph of the Introduction. We have added a statement with this caveat in the second and third paragraphs of the Discussion.

The nanobody could potentially drive conformational remodeling of the receptor. However, Nb39 does not bind to receptor in the absence of ligand so is not driving an active conformation by itself and therefore not forcing a ligand to bind to a certain conformation. It is therefore likely that the agonist must first stabilize the receptor in an “active-like” ensemble that is competent to bind Nb39, thus the Nb39 on-rate should be directly related to the agonist’s ability to produce these active-like states. We have added a statement in the ninth paragraph of the Discussion.

3) Since µ-PAMs alone promote Nb39 binding but show no signaling is it accurate to define this assay as one measuring efficacy/intrinsic activity?

Although the mu-PAMS do not activate G protein, as measured by the [^35^S]GTPγS assay, they do inhibit adenylate cyclase activity, but this is very poor in line with their weak ability to recruit Nb39. Thus, we do believe this assay is measuring efficacy, and that the sensitivity of this assay is high and on the order of assays measuring downstream (and amplified) second messengers. This is discussed in more detail in our answer to point 5, below.

4) The evidence for ligand-induced unique/distinct conformations is not clear. There are kinetics of ligand association/dissociation, ion interactions, as well as the nanobody associations and dissociations which are the measured output. Without knowing the kinetics of all players how can alternative conformations be assumed as opposed to duration spent in common conformational states?

The point we make is that nanobody dissociation is altered differentially when the receptor is bound to certain agonists, and we suggest that these agonists possess unique cooperativity with Nb39 binding that allows them to produce an altered rate of dissociation of Nb39. We agree that the kinetics of ligand and ions etc. could be a confounding factor but it is clear from our data that most compounds follow a similar pattern of Nb39 dissociation except for one or two compounds, and, for example, it is not related to ligand efficacy. If ligand dissociation is a confounding factor then we might have expected differences across all ligands. However, this is not the case; the dissociation rate constant for Nb39 was similar in the presence of quickly-dissociating agonists like the endogenous ligands and slowly-dissociating agonists including buprenorphine. Nonetheless, we have added a caveat in the Discussion (eighth paragraph).

5) In many PAM programs, some of the members are PAM Agonists, i.e. it is a fine line between making it easier for an agonist to promote an active state vs. actually having the PAM itself induce the active state. It would be nice if the authors expanded on their thoughts on this a little more.

We agree with this and our data do show that the modulators themselves have some intrinsic activity because they recruit Nb39 to the receptor, albeit to a low extent (see also response to point above). This agrees with our recently published data showing that BMS986187 has agonist activity at mu-receptors at the level of adenylate cyclase (Livingston et al., 2018), although the level of efficacy is too low to be observed in GTPγS or β-arrestin assays. The high sensitivity of the assay is an additional strength. We have proposed (Livingston and Traynor, 2014) that PAM activity at the mu-opioid receptor is due to a negative, indirect, interaction between the PAM site and the Na^+^ ion site such that this makes it more favorable for an orthosteric agonist to drive formation of R* – this would be a PAM. If the PAM alone is sufficient to drive Na^+^ from its binding site this would favor transition to R* and so be an ago-PAM. Thus, there is a continuum between a NAM (negative modulator), SAM (silent modulator) which shows no interaction with the Na^+^ site, a PAM and an ago-PAM. The sixth paragraph of the Discussion has been extended to include these comments.

6) It is not clear exactly which 'allosteric binding model' is used in this study – standard Graphpad model? Is this the Hall model or more comprehensive model? This should be specified. Also, the authors discuss 'allosteric efficacy' and this is confusing since this implies beta effects from the allosteric functional model whereas it appears only alpha effects where measured – alpha is only part of the story as beta effects can increase or decrease ternary and quaternary complex production. In this regard, the authors need to specify why only affinity measurements should reflect everything there might be to know about the quality of the active state ensemble.

This refers to data in Figure 5D reported using the standard GraphPad model, so we agree we are only measuring alpha effects. This is now specified in the first paragraph of the subsection “Allosteric modulation of µ-OR in rHDL by small-molecule PAMs”. We used the term “allosteric efficacy” as a general term to describe the enhancement of any agonist-dependent parameter. We have replaced the words allosteric efficacy in the second paragraph of the subsection “Allosteric modulation of µ-OR in rHDL by small-molecule PAMs” and removed these words from the section heading.

7) In general, more discussion of when only G protein alpha effects reflect total opioid agonist efficacy would make this a stronger paper.

Much of the study uses Nb39 but we do show that this also applies to heterotrimeric Gabg proteins. For each Gα activated there is one Gβγ subunit so that the rank order of efficacy for measures downstream of heterotrimeric G protein should not change. However, we agree we are only measuring Gα effects and this is a limitation when considering all of the downstream processes that could involve Gα and Gβγ (e.g. Adenylate cyclase for Gα; GIRK for Gβγ) with different efficacy requirements. On the other hand, the method is a way to directly compare efficacy of the opioid molecules not the many different cascades that can happen downstream of the receptor. A comment has been added to the fifth paragraph of the Discussion.